# Characterization of Tailings Dams by Electrical Geophysical Methods (ERT, IP): Federico Mine (La Carolina, Southeastern Spain)

Julián Martínez [1,*] , Rosendo Mendoza [1] , Javier Rey [2] , Senén Sandoval [3] and M. Carmen Hidalgo [2]

1    Departamento de Ingeniería Mecánica y Minera, EPS de Linares y CEACTEMA, Universidad de Jaén, Campus Científico Tecnológico, Linares, 23700 Jaén, Spain; rmendoza@ujaen.es
2    Departamento de Geología, EPS de Linares y CEACTEMA, Universidad de Jaén, Campus Científico Tecnológico, Linares, 23700 Jaén, Spain; jrey@ujaen.es (J.R.); chidalgo@ujaen.es (M.C.H.)
3    Everest Geophysics SL, Colmenarejo, 28270 Madrid, Spain; senen@everestgeophysics.com
*    Correspondence: jmartine@ujaen.es

**Abstract:** This work analyzed the effectiveness of two electrical geophysical methods in characterizing tailings dams. A large flotation cell used for sludge thickening in the concentration plant of the Federico Mine (closed in 1985) within the old mining district of La Carolina (southeastern Spain) was selected for this research. In addition to the direct information provided by the geology of the study area and the surface exposure of the waste deposits, information regarding the construction of this mining structure was available, which helped in the interpretations of the geophysical survey data. In this study, two geophysical surveying methods were used simultaneously: Electrical resistivity tomography (ERT) and induced polarization (IP). Six profiles were acquired, processed, and interpreted. The length of the profiles allowed the obtaining of data reaching maximum investigation depths ranging between 7 and 65 m. These profiles provided information for a detailed analysis of the internal characteristics of the deposited materials. The lateral and vertical observed variations are linked to different degrees of moisture content. The study also defines the geometry of the top of the bedrock and the tectonics that affect the pouring/dumping hole. Old flotation sludge has resistivity values that range between 1 and 100 $\Omega$m (i.e., wet waste 1–30 $\Omega$m, dry waste 30–100 $\Omega$m), while phyllites in the rocky substrate have resistivities larger than 200 $\Omega$m and can even reach va-lues greater than 1000 $\Omega$m. Between the waste and unaltered phyllites, there is a supergene alteration zone (weathering) with resistivity values between 100 and 200 $\Omega$m. The IP method was used to detect the presence of metals in the accumulated waste in the pond by analyzing the presence of large chargeability anomalies. Anomalies were detected in four of the profiles, which ranged from low (i.e., between 0 and 8 mV/V) to medium (i.e., between 8 and 18 mV/V) and high values (i.e., 18 and >30 mV/V).

**Keywords:** tailings dam; electrical resistivity tomography; induced polarization; La Carolina Mining district; Spain

## 1. Introduction

Mineralized veins in the mining district of Linares and La Carolina (SE Spain) are enriched in metallic sulfides, mainly galena (PbS), and have been exploited by underground mining since pre-Roman times until the third quarter of the 20th century, when the last mine closed. The materials extracted from the mine were subjected to a mineral concentration process in treatment plants located nearby. Gravimetric methods were used until the 1950s. From then onward, flotation techniques became the preferred method causing accumulations of generated wastes in tailings ponds and dams [1,2].

The metal crisis of the 1980s, along with the depletion of some deposits, caused mining company closures and the dismantling of their facilities, leaving waste on the ground

without the necessary corrective measures being taken to avoid mobilization of the metals contained in the waste. These conditions pose an environmental hazard, due to the lack of preventive measures in the design (conditioning and waterproofing of the site). Failure to seal and revegetate these abandoned sites have exposed the structures to the action of atmospheric agents [3–7]. Mobilization of the metal(oid)s present in such wastes affects the surrounding soils and channels, as has been shown, for example, in Goseong (Korea), where high levels of Cu, Pb, As, and Zn have been measured and detected in crops [8].

In the Linares-La Carolina mining district, there are 33 tailings dams that accumulate 10,500,000 m$^3$ of waste. This amount is extremely high, especially considering that in the Region of Andalusia, 105 tailings dams have been inventoried out of a total of 344 metal ore processing dams in Spain [9].

Currently, any remediation effort to be carried out requires detailed knowledge of the characteristics of the accumulated waste. Good knowledge is needed about its limits and variations both in depth and laterally and its relationship with the underlying substrate to recognize possible leachate leakage areas, both at the foot of the dam and by direct infiltration into the ground affecting the groundwater system.

Geophysical methods have been successfully used in mining research: Gravimetry [10,11], magnetometry [10,11], electrical [12], electromagnetic [13], and seismic [14,15]. In recent years, these techniques have been applied to evaluate and resolve environmental problems associated with mining activity and the generated waste. In Spain, the electrical resistivity tomography (ERT) technique has been used in a variety of mining districts. For example, in Cartagena (SE Spain), multiple studies have recently evaluated the mobility of metals in tailings dams [16,17], using a combination of ERT, mechanical drilling, and geochemical techniques [18]. In the mining districts of Linares and La Carolina (southern Spain), different tailings ponds and slag heaps have been characterized by applying all of these techniques [3–7]. Similarly, in the Iberian Pyrite Belt (SW Spain), ERT has been used to evaluate tailings ponds created during the concentration of massive sulfides and the generation of acidic waters [19–21].

These methodologies have also been used in other mining districts worldwide. Such is the case in Kettara (Morocco), where electrical geophysical methods have been applied to study a sludge pond created from treating pyrrhotite [22]. In the Zaruma-Portovelo region (Ecuador), ERT has been applied to study the proposed site of a gold mining sludge pond, which was rejected because of possible landslide problems in the granites where this pond would have been located [23]. Additionally, in the Haveri area of Finland, a combination of geophysical and geochemical techniques has been used to evaluate the generation of acidic waters from an Au-Cu mine tailings pond [24].

## 2. Description of the Study Area

### 2.1. Geological Setting

The study area is located in the municipality of La Carolina, Jaén (Figure 1a), in the Sierra Monera mountain massif, specifically on the southeastern slope, which is the southeastern limit of the Hesperian massif [25,26]. Two large geological units are distinguished regionally: The Palaeozoic basement (Figure 1c) and post-Hercynian cover.

The Palaeozoic basement consists of a metasedimentary series composed of phyllites and quartzites from the Ordovician to the Carboniferous [27–29], and a granitic intrusion was emplaced in this area at the end of the Hercynian orogeny [26].

Subsequent to this intrusion, an extensional episode occurred that favored hydrothermal fluid circulation though a network of fractures, which gave rise to the Philonian reservoir in La Carolina. The veins exhibit mineral paragenesis that mainly consists of sulfides with galena, chalcopyrite, sphalerite, and pyrite predominating, accompanied by quartz, ankerite, and calcite [29].

On the Palaeozoic basement, the post-Hercynian series is horizontally positioned and consists of Triassic, Miocene (not present in the work zone), and Quaternary materials, which sometimes hide the discordantly arranged mineralized zones.

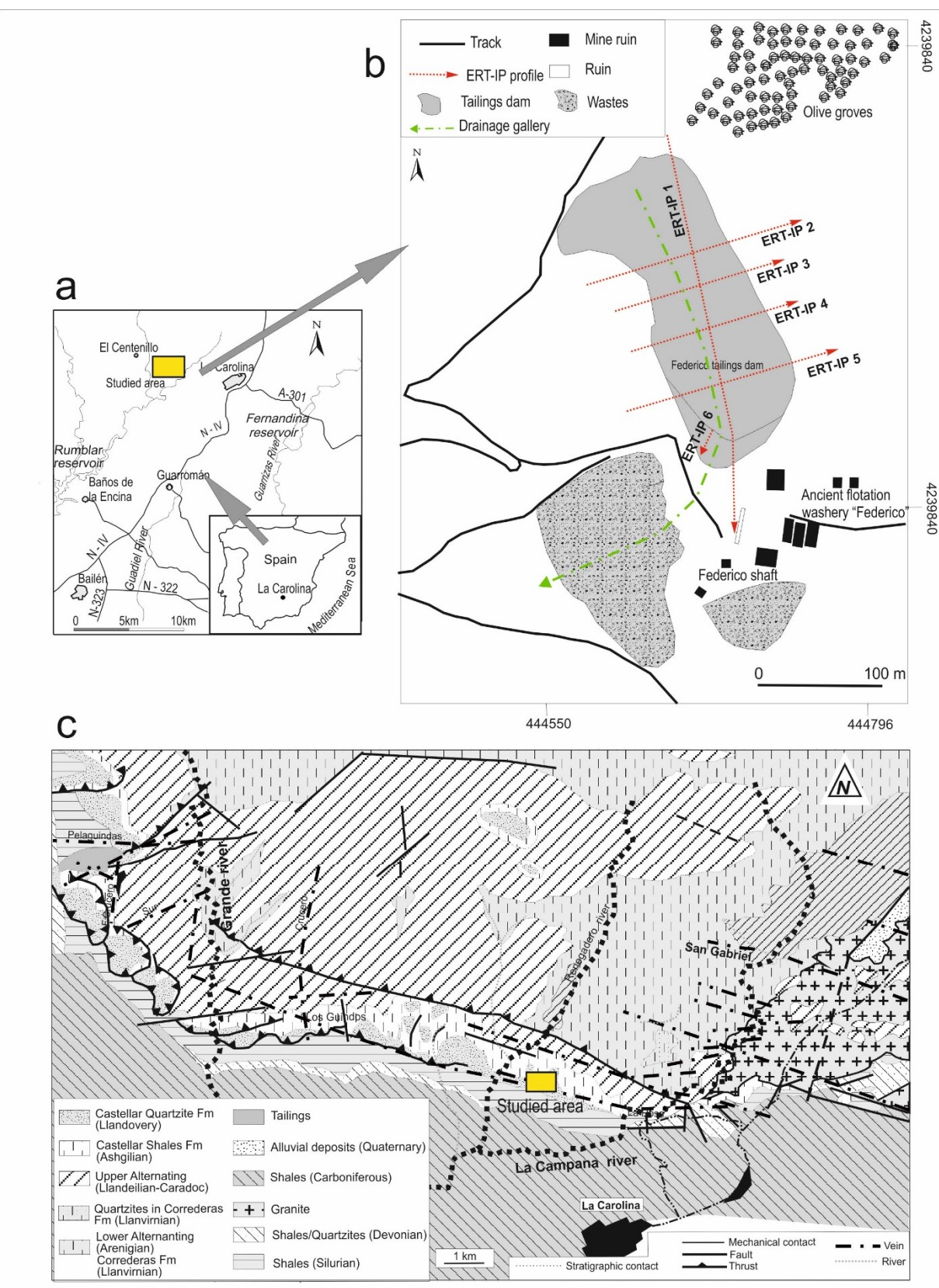

**Figure 1.** (**a**) Location of the Federico mine dam (white box). (**b**) Federico Dam and location of the investigation profiles.
(**c**) Geological cartography (based on the work of Castelló and Orviz in 1976) showing the position of the studied sector.



### 2.2. Mining Technique

Underground mining methods were used in this mining district [1] and included shrinkage stoping and cut-and-fill stoping. These methods were the most common methods for the Philonian deposits that presented subvertical veins in hard embedded rocks (granites) and were the second-most common used for the mineralization that occurred more in the upper part of the district and were sometimes disseminated (phyllites). The shrinkage stoping technique consists of stripping minerals by drilling and blasting through ascending horizontal strips from the bottom up. The overturned minerals are left in the stope to serve as a working platform and as a provisional support for the gables until it is finally abandoned once the chamber has been excavated [30]. In the cut-and-fill stoping method, the void created by extracting the ore along a horizontal strip over the seam is filled with loose material from waste rock, which can sometimes come from outside the mine and serves as a support for the host rock and thus ascends during chamber excavation.

### 2.3. Tailings Dam

The Federico mine dam (Figures 1 and 2) is located at a site with the same name, with UTM coordinates 444.900/4.239.700 and a height of 620 m. This tailing dam contains 600,000 $m^3$ of slate and quartzite sludge.

The mineral concentration that took place in the La Carolina district was structured in two stages. The stripped materials (with high grades of 18% to 20% galena after a previous separation operation in the mine) were subjected to a gravimetric process from which, on the one hand, the first galena concentrate was obtained and, on the other hand, some mixed and high-grade wastes were then treated by a second flotation process. The flotation method required milling the mixture to sizes below 1 mm (approximately 50–70 μm) to release the minerals, and a fine concentrate of galena, as well as a residue that was pumped into ponds and tailings dams.

These structures, as in the case of the Federico mine (Figure 1b), were built using a depression in the ground where a gravel retaining wall was available to form the dam. The flotation waste was pumped from the plant by discharging through a gutter placed at the top of the dam through multiple outlets that were opened or closed to regulate the deposition of the discharged sludge. The accumulated mud formed a beach area next to the wall and a lake of process water. Water level management was carried out either through a central chimney connected to a bottom drainage gallery (in ponds on horizontal terrain) or, as in the case of Federico, through a lower drainage gallery built of masonry on a gravel base, to which a drainage tube was connected through which the process water was freely discharged to take it out of the structure [18] (Figure 2a–d).

In other similar structures of the La Carolina district, the waste metalloid contents have been analyzed. Such is the case for the Aquisgrana sludge dam, where concentrations of 544, 987, 8065, and 5186 mg/kg were obtained for As, Mn, Pb, and Zn, respectively [31].

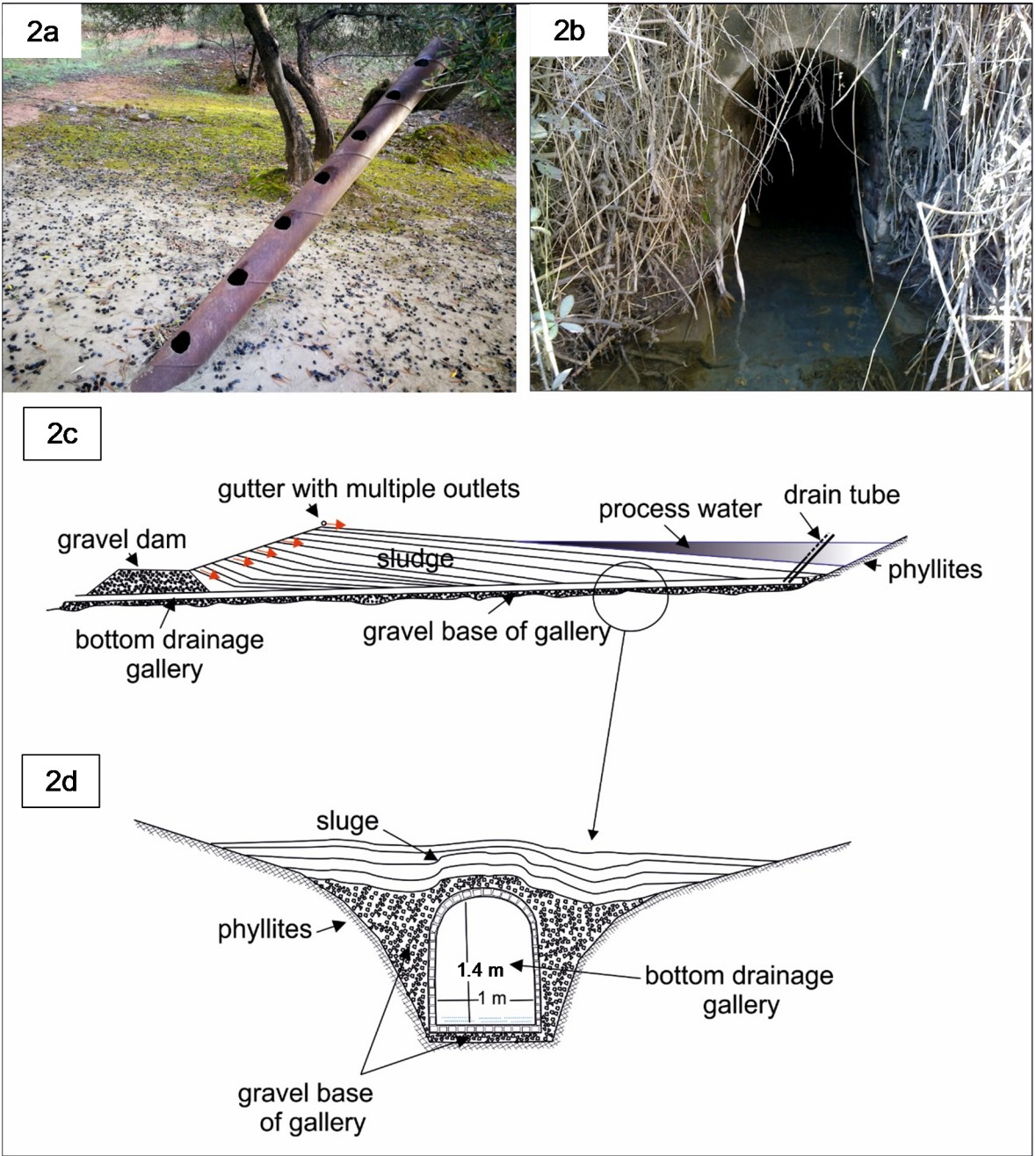

**Figure 2.** (**a**) Drainpipe connected to the bottom drainage gallery of the Federico dam. (**b**) Bottom discharge-type gallery of a tailings dam. (**c**) Construction diagram of the Federico flotation tailing dam. (**d**) Details of the bottom drainage gallery of the dam, which was constructed with masonry and covered with gravel.

## 3. Materials and Methods

*Electrical Resistivity Tomography (ERT) and Induced Polarization (IP)*

Ground resistivity measurements can be made with either an alternating current (AC) or a direct current (DC). Direct current (DC) resistivity methods use a controlled source of electrical current to produce an electrical voltage in the ground. Current is introduced into the ground through a pair of electrodes and the potential field is measured using another pair of electrodes.

The ERT technique can be considered an evolution (or combination) of other well-stablished geoelectrical techniques, such as vertical electrical soundings (VES) and the

constant separation traversing (CST). It is a noninvasive technique that produces nume-rical models of the subsurface representing the 2D or 3D distribution of electrical resisti-vity in the ground [32–35]. The technique consists of acquiring large datasets of apparent resistivity in the field, filtering them, and finding a resistivity distribution of the ground that produces a similar response to that measured in the field. The models are derived by an iterative inversion process.

In recent years, this technique has expanded rapidly with the development of auto-mated data acquisition systems capable of handling large amounts of data that are quickly processed with appropriate tools and that facilitate the generation of increasingly complex geological models [36].

In the data collection process, numerous electrodes are placed along a profile, with separations determined as a function of the degree of resolution and depths to be inves-tigated. Thus, the smaller the separation between electrodes, the greater the resolution obtained and the lower the depth investigated, while increasing interelectrode separations enable deeper depths of investigation but with lower resolution [33]. We have chosen the Wenner–Schlumberger arrangement, which has a good penetration capacity and a good sensitivity for both vertical and horizontal structures. Other electrode arrangements such as the Wenner or dipole–dipole configurations, are more appropriate for sub-horizontal structures and vertical structures, respectively.

The technique is being used with good results in different earth science fields: Stratig-raphy [6,37,38], hydrogeology [39], environment [7,16,17,20,38], and mining [40,41].

The induced polarization (IP) method is used to measure the electrical charging capacity of a given body or material in the subsoil and can detect those materials that have the ability to store a current for a given time [32,42]. To apply this method, a known direct current is applied between two electrodes (current electrodes) for a predetermined period of time (usually longer than one second). When the current is abruptly terminated, the voltage decay between another pair of electrodes (electrode potentials) is measured. A parameter called apparent chargeability can be estimated by measuring the area under the voltage decay curve. This value is normalized by the normal potential, Vo, which is obtained during the period that the known voltage is applied to the current electrodes [43].

The equipment used in this work was a digital resistivity meter RESECS manufactured by Deutsche Montan Technologie (DMT) (Figure 3a), which uses RESECS32 software for field data acquisition. The data sequence is selected to measure the ground under the profile with a dense cloud of measurement points. At each point, two parameters are determined: The apparent electrical resistivity (measured in $\Omega$m) and the apparent charge-ability (measured in ms or mV/V). The resistivity meter controls the current application and measures voltages using nonpolarizing electrodes with spiral lead cores immersed in a saturated lead chlorite gel (Figure 3b).

In this work, six ERT resistivity and IP profiles were acquired (Figure 1b) with a Wenner–Schlumberger configuration. For each profile, 64 electrodes were laid out. The lengths of the profiles were 320 m for ERT-IP 1, 192 m for ERT-IP 2, 128 m for ERT-IP 3, 128 m for ERT-IP 4, 192 m for ERT-IP 5, and 36 m for ERT-IP 6. The electrode separation "a" was 5, 3, 2, 2, 3, and 0.5 m, respectively (Figure 3c, profile ERT-IP 6). The parameters used to measure the resistivity and the chargeability were as follows: Applied voltage = 800 V; time of current injection = 1024 ms (with a delay of 10 ms to allow for stabilization and voltage measurement); interval between measurements of 990 ms (where the voltage decay curves were measured); no filters; and one repetition cycle (in profiles 1 to 5). In profile ERT-IP 6, an 880 V current was used; time of application = 2000 ms (with a delay of 50 ms) and interval of 1000 ms (for voltage decay readings); a Bessel filter and two repetition cycles.

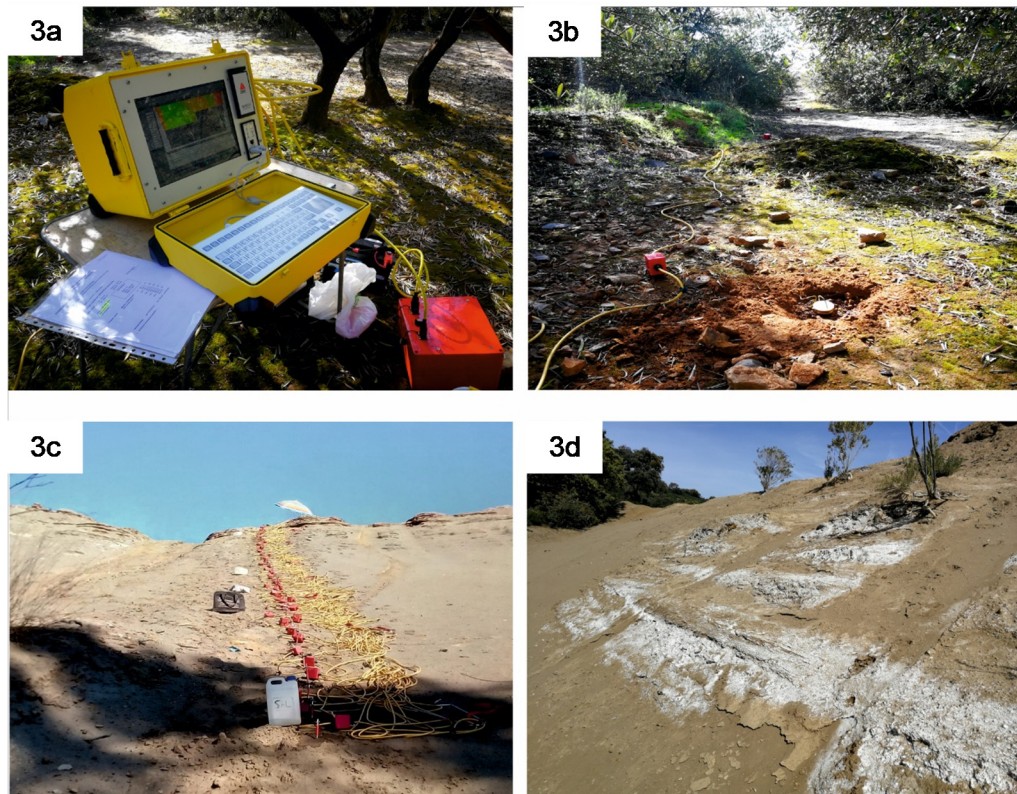

**Figure 3.** (**a**) Deutsche Montan Technologie (DMT) RESECS resistivity meter. (**b**) Nonpolarizable electrodes in the profile ERT-1. (**c**) Profile ERT-6 on the dam slope. (**d**) Crusts observed on the dam slope.

To generate the resistivity and chargeability values, RES2DINV software (Loke and Barker, 1996) was used, which uses the least squares method with forced smoothing and a modified quasi-Newton optimization technique, in which a subsoil model is constructed using rectangular prisms. The resistivity values for each prism are determined and the differences between the observed and calculated apparent resistivity values are minimized [44–46]. The field apparent resistivity and apparent chargeability values were previously processed and filtered with PROSYSII software (IRIS Instruments). At this stage, the topographic information (topographic variations along the profile) was incorporated into the data.

## 4. Discussion and Results

*Electrical Resistivity Tomography (ERT) and Induced Polarization (IP)*

Six ERT and IP profiles were acquired (Figure 1b) on a tailings dam in the abandoned Federico Mine. One was longitudinally oriented (ERT-IP 1) along the major axis of the tailings dam. Four were transversely oriented (minor axis of the tailings dam, ERT-IP 2, 3, 4, and 5). The last one was oriented along the maximum slope of the dam (ERT-IP 6). Figures 4–6 show the derived resistivity and chargeability models.

The results of the study allow the geometry of the top of the bedrock basement to be defined and provides information about the tectonic fractures. The flotation sludge has resistivity values that range between 1 and 100 Ωm (wet waste 1–30 Ωm, dry waste 30–100 Ωm), while the phyllites of the rocky substrate have resistivities larger than 200 Ωm and can even reach values over 1000 Ωm. Between the waste and the unaltered phyllites, there is a supergene alteration zone (weathering) with resistivity values between 100 and 200 Ωm.

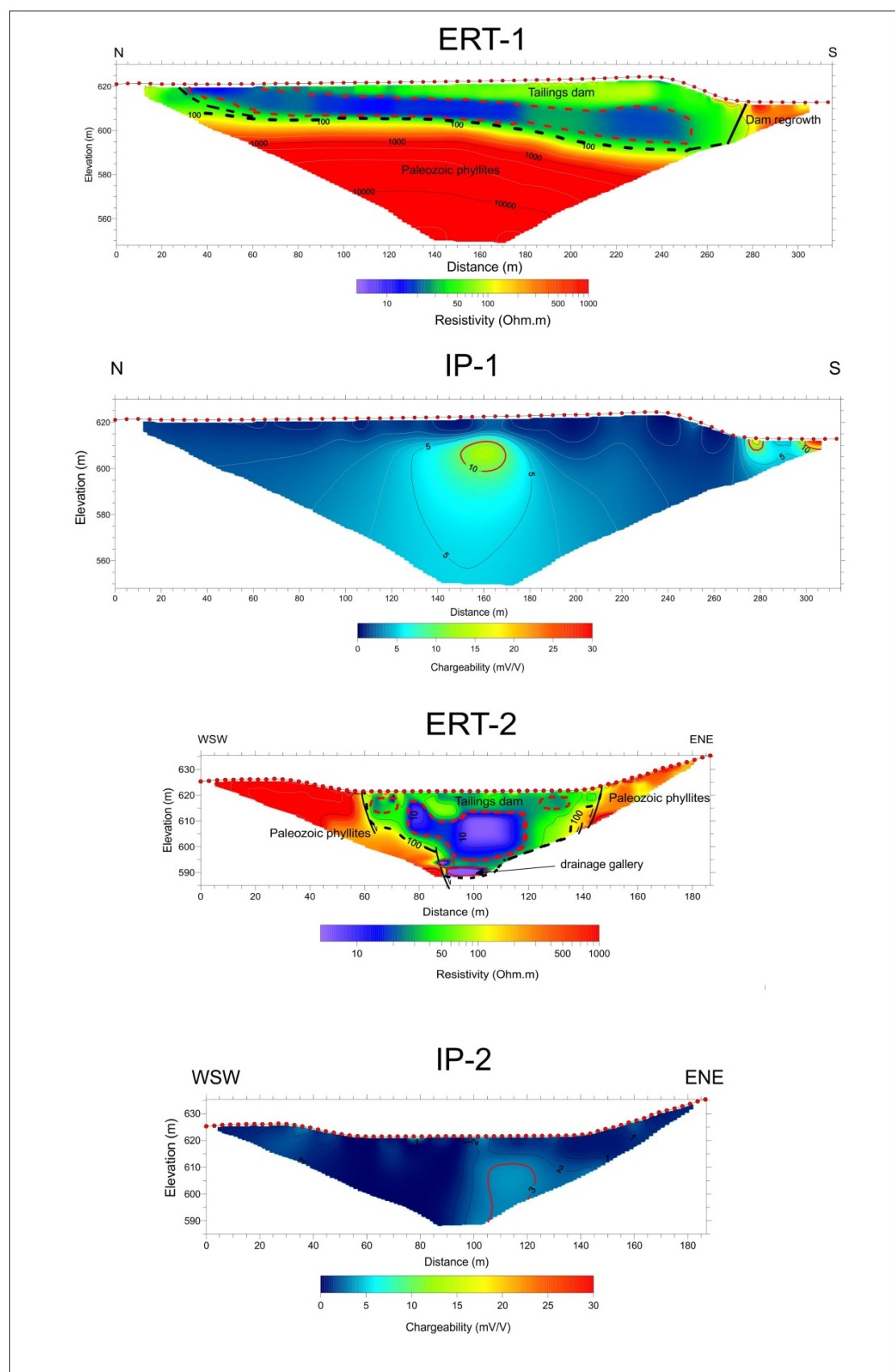

**Figure 4.** Resistivity models ERT-1 and ERT-2, which show the underlying Palaeozoic (phyllites) with high resistivity values (red colors), and flotation waste with low resistivities (blue and green colors) that indicate wet areas (dashed red line). In induced polarization (IP) models IP-1 and IP-2, low chargeability values are shown as blue and yellow colors and red are associated with the level enriched in metal(oid)s.

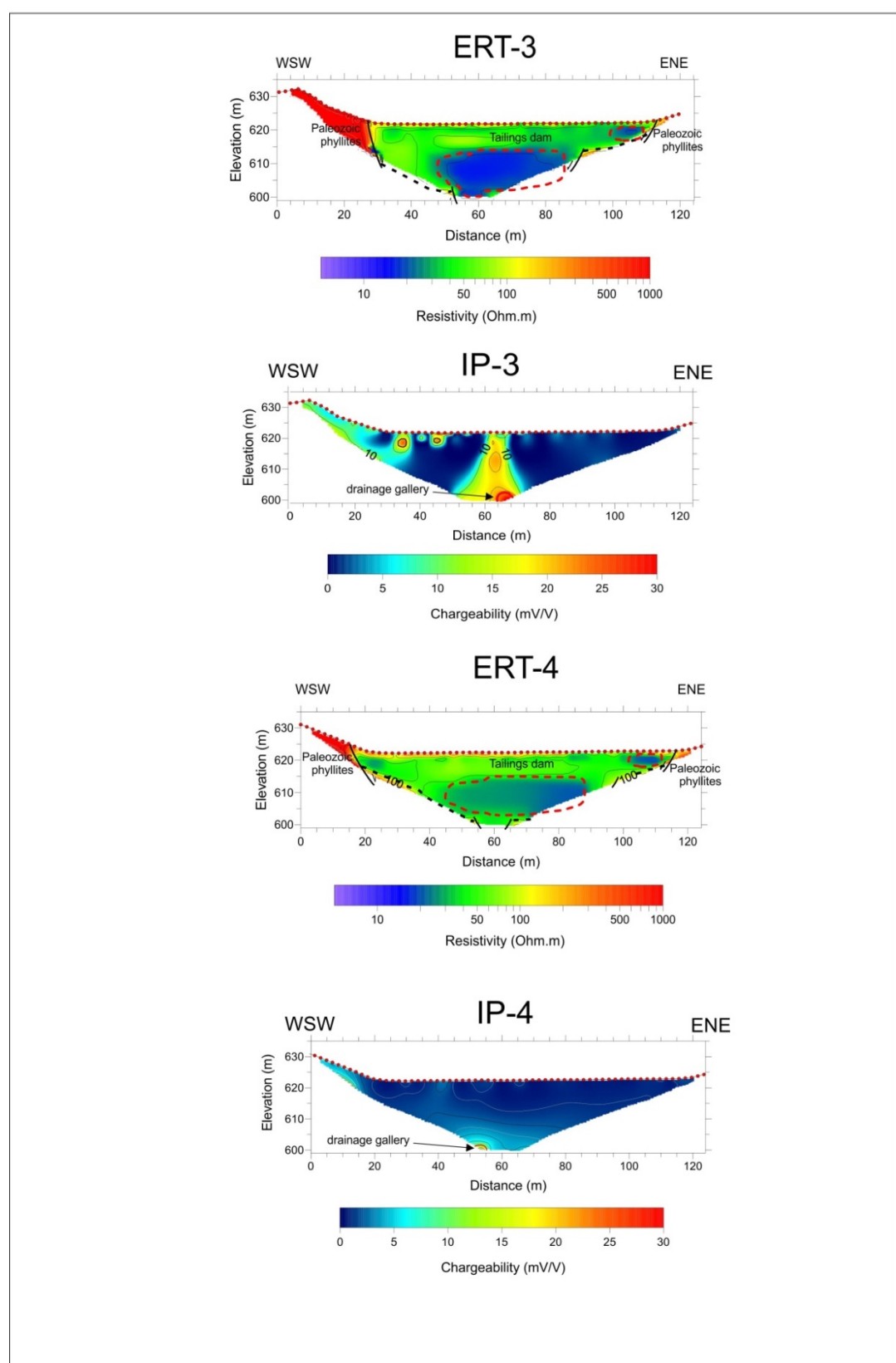

**Figure 5.** Resistivity models ERT-3 and ERT-4, in which the underying Palaeozoic (phyllites) with high resistivity values (red colors) on the sides of the profiles and flotation waste with low resistivities (blue and green colors), are observed, which indicate wet areas (dashed red line). In IP models IP-3 and IP-4, chargeability values are shown in blue for the sluge. The yellow- and red-color anomalies in IP 3 and IP 4 are associated with the bottom drainage gallery.

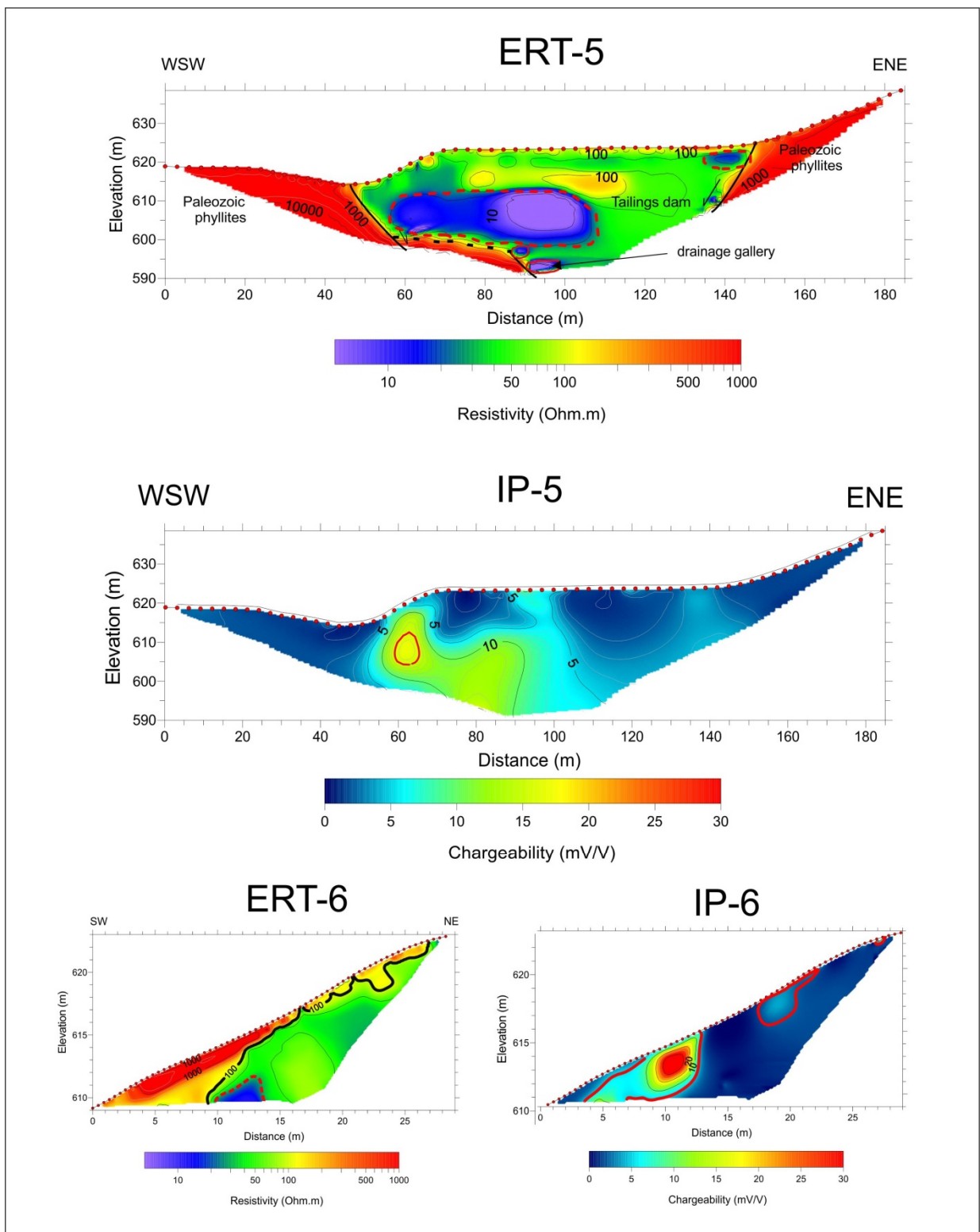

**Figure 6.** Resistivity models ERT-5 and ERT-6, the underlying Palaeozoic (phyllites) with high resistivity values (red co-lors) on the sides of the profiles, and flotation waste with low resistivities (blue and green colors) are observed, which indicate wet areas (dashed red line). ERT-6 (dam slope) exhibits a different response to the waste, with low values (in blue and green) in areas with specific moisture levels. Yellow colors are associated with crust. In IP model IP-5, low chargeability values are observed, with the medium chargeability zone associated with wet sludge. IP-6 exhibits a strong chargea-bility anomaly probably associated with levels of disseminated sulphides and metal(loid)s-enriched crust.

In the six analyzed profiles, IP measurements were also conducted. The apparent chargeability measurements were then inverted to obtain subsurface chargeability models (Figures 4–6). The inversion results differentiate three ranges of chargeability: Low (0–8 mV/V), medium (8–18 mV/V), and high/very high (18–30 mV/V). Low chargeability va-lues are probably associated with the absence of dispersed metals. They are characterized by a weak polarization phenomenon leading to the low observed chargeability values.

The ERT and IP 1 profile (Figure 4) has a length of 320 m in the N–S direction and reaches a depth of 64 m. The final RMS error (for the simultaneous inversion of resistivity and chargeability data) is 5.52%. The obtained resistivity distribution shows two distinct units: The deepest is characterized by high resistivity values (between 200 and more than 1000 Ωm) that correspond to the Palaeozoic substrate, which is composed mainly by phyllites with quartzite intercalations. These materials are at the base of the profile, are uniform, and exhibit a slight slope toward the South. Overlying these materials, resistivities between 100 and 200 Ωm are found. They are associated with the weathering of the phyllites. At the surface, flotation waste lays directly over these materials with resistivities below 100 Ωm. They are composed by silts and fine sands with thicknesses up to 35 m toward the S, adjacent to the dam wall. Within this waste, two different electrical responses are observed: A zone with resistivities between 1 and 30 Ωm, where we assume that the waste contains a high amount of moisture; and values between 30 and 100 Ωm, which are characteristic of dry waste. The resistive materials observed at the southern end of the profile correspond to the mine gravels used to form the dam wall.

The IP 1 model (Figure 4) shows generally low chargeability values (0 to 8 mV/V). A moderate-amplitude anomaly (8–18 mV/V) appears centered inside the waste body associated with a possible metallic sulfide-enriched zone. At the southern end, two maxima are observed that could be related to objects buried in the gravel dam wall.

The ERT and IP 2 profile (Figure 4) has a length of 180 m and reaches a depth of 36 m. The final RMS error is 1.60%. As in the previous profile, the phyllitic basement waste is clearly differentiated from the bedrock. This waste can also be described according to the resistivity values influenced by the moisture content. A very low-resistivity anomaly surrounded by more resistive (less humid) sludge appears in the center of the model. It should be noted that the low-resistivity values present in the lower part of the deposit (9.5 Ωm) are associated with the bottom drainage gallery on a gravel base (Figure 2). The deeper phyllites with high resistivities were reworked, creating a set of fractures asso-ciated with the trough that serves as the discharge hole.

The IP 2 profile (Figure 4) shows a dominating low-chargeability distribution without any remarkable anomaly.

The ERT and IP 3 model (Figure 5) has a length of 120 m and reaches a depth of 24 m. The final RMS error is 2.53%. A wet core (low resistivities) is present within the waste. There is another conductive zone toward the ENE, at the edge of the model. Given that the wet core is located at the end of the dam, it may be associated with leaching zones. This model also shows the possible presence of the bottom drainage gallery with low resistivity values (Figure 2). It should be noted that in all models, an increase in resistivity (between 80 and 150 Ωm) is observed within the dry zone of the waste, which we associate with the formation of crusts that have resulted from changes in the characteristics of the deposited slurries that facilitated oxidation (Figure 3d). At the sides of the model, Palaeozoic resistive phyllites are observed, probably associated with a fracture-type contact.

The IP-3 model (Figure 5) has an anomaly with high chargeability values located in the center of the waste extending toward the bottom (although it seems to project from the bottom upward). This anomaly is associated with the presence of the bottom drainage gallery and filling gravels that surround the gallery (Figure 3a,b). This can also represent an area that could be significantly enriched in metals at the initial location of the discharge of the flotation fines. The two small peaks observed toward the WSW can also be asso-ciated with metal-enriched areas.

The ERT and IP 4 model (Figure 5) has a length of 120 m and reaches a depth of 24 m. The final RMS error is 8.64%. This profile exhibits the same electrical pattern, with a wet core of low resistivity (<20 Ωm) enveloped by dry wastes (20–100 Ωm). The most conductive zone, which is toward the ENE end of the profile, is observed close to the phyllite contact. It could be a possible water circulation zone within the waste. The higher resistivity values that we associate with crusts are also noteworthy, which in this profile are clearly manifested on the dam surface (Figure 3d). Phyllites (high resistivities) are detected at both ends of the profile.

The IP-4 model (Figure 5) exhibits low chargeability values. A small anomaly in the lower part of the model is associated with the bottom drain.

The ERT and IP 5 model (Figure 6) has a length of 180 m and reaches a depth of 36 m. The final RMS error is 10.93%. It presents a resistivity distribution similar to those of the other profiles and clearly differentiates the phyllites (resistivities greater than 200 Ωm) from the flotation waste (resistivities less than 200 Ωm). The phyllites are interrupted by a series of fractures that constitute the trough. Several high-resistivity zones are observed in the central core and in the lateral zone toward the ENE and in the bottom gallery (supported and surrounded by gravels, Figure 2) within the waste depending on the moisture levels. It should also be noted that the resistivities between 80 and 150 Ωm obtained in the waste are associated with crust formation, which in this profile are observed both inside and on the surface (Figure 3d).

The IP 5 model (Figure 6) exhibits two patterns in the chargeability distribution. Low values characterize both the waste and phyllites. On the other hand, high values between 8 and 18 mV/V are observed both toward the WSW and in the background of the waste. We associate these anomalies with zones enriched in metallic sulfides and wet waste.

In addition to the longitudinal and cross-sectional profiles, a high-density profile (ERT and IP 6; Figure 3c, Figure 6) was aquired with an electrode spacing of 0.5 m on the slope of the dam to study the lateral behavior of the waste. The RMS error is 5.07%. Two very sharp resistivity changes are observed. One is associated with the contact between the unaltered waste (5 to 100 Ωm) and the wet core (5 to 20 Ωm). The second is the contact between high resistivities associated with the crusts observed inside and on the surface of the waste (Figure 3d).

The detailed model, IP 6 (Figure 6), exhibits low background chargeability values and a zone with high and very high values (8–30 mV/V) on the surface of the lower part of the dam slope. The latter can be associated with a local concentration of metal sulfides.

## 5. Conclusions

In this work, two electrical geophysical methods (ERT and IP) were used in an integrated manner to analyze their capability to model the inner structure of tailings dams. For this study, the abandoned dam at the Federico mine (La Carolina, southeastern Spain) was chosen. It contains flotation waste from an old washing plant, one of the most important in the mining district. The ERT and IP methods were used and their results were compared with direct field observations and other available information regarding the mining structure.

In the electrical tomography models, the mining wastes of the base rock (Palaeozoic phyllites) are clearly defined. The method allows us to deduce the morphology of the phyllitic substrate, which is controlled by the presence of fractures and which forms a trough that serves as a repository for flotation wastes. ERT also allowed us to interpret changes in the interior of the waste. Wet and dry areas were detected, in addition to crusts and levels associated with higher metal sulfide contents. The presence of dam drainage at the bottom of the dam is also detected with this method.

On the other hand, the IP method complements the information obtained with ERT and shows three chargeability levels associated with the bottom of the waste, levels with sulfide enrichment associated with crusts, and a level associated with the bottom drainage gallery.

These indirect electrical methods are of interest for examining tailings ponds and dams, as they allow the definition of both their morphology and deposition on the ground, as well as their internal structures, and constitute a step that would be conducted prior to conducting mechanical soundings.

**Author Contributions:** Conceptualization and investigation, J.M., R.M. and J.R; software, J.M. and S.S; writing and review, J.M., S.S. and M.C.H. All authors have read and agreed the published version of the manuscript.

**Funding:** This research received no external funding.

**Institutional Review Board Statement:** Not applicable.

**Informed Consent Statement:** Not applicable.

**Data Availability Statement:** Please refer to suggested Data Availability Statements in section "MDPI Research Data Policies" at https://www.mdpi.com/ethics.

**Acknowledgments:** Careful reviews by Mario Zarroca, Roberto Rodríguez and Isaac Corral and anonymous referees by improved the manuscript.

**Conflicts of Interest:** The authors declare no conflict of interest.

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
