# Peer review of "Characterization of Tailings Dams by Electrical Geophysical Methods (ERT, IP): Federico Mine (La Carolina, Southeastern Spain)"

_minerals, doi:10.3390/min11020145_

Round 1
Reviewer 1 Report
The case study is interesting and developed by a group of specialists in the study of mining tailings, with several relevant publications. In the present study, the authors make a great confusion between method and technique geophysical, something pointed out in a recurrent way throughout the text, where several suggestions for alteration were inserted. We recommend consulting textbooks in geophysics. The authors need to describe in more detail the DC resistivity method and the electrical tomography technique, independently. Inversion models need to display RMS values. We recommend a joint discussion of resistivity and chargeability for each ERT line, because the form presented in the text discusses the IP data after the presentation of the inversion models and makes the text confusing

Author Response
(Response to comments made in the pdf) (In color red)
Tittle
I have modified the title partially. Characterization of tailings dams by electrical geophysical methods techniques (ERT, IP): Federico Mine (La Carolina, Southeastern Spain)
Line 3: I agree with the referee, techniques I have changed for methods.
Abstract
Line 12: Techniques I have been changed for methods.
Line 24: I have changed technique by method.
Introduction
Line 62: I have added… and affecting the groundwater system.
Line 63: I've changed …. techniques by methods
Line 77: I've changed …. techniques by methods
Description of the Study Area
Line 90: In the Figure 1C, I have increased the size of the characters.
Materials and Methods
Line 150- …: Brief explanation of the DC method and the ERT as required by reviewer.
Line 165..: I have added… We have chosen the Wenner-Schlumberger arrangement which has a good penetration capacity and a good sensitivity for both vertical and horizontal structures. Other electrode arrangements such as the Wenner or dipole-dipole configurations are more appropriate for sub-horizontal structures and vertical structures, respectively.
Line 186: I have modified the figure caption
Discussion and Results
Line 204: I modified the point 4.1. by Electrical Resistivity Tomography (ERT) and Induced Polarization (IP)
Line 207-209: I have changed the description of the paragraph
Line 212: I have added the RMS
Line 224: I have added the RMS
Line 237: I have added the RMS
Line 248: I have added the RMS
Line 262: I have added the RMS
Line 266: I have added the RMS
Line 282: I have added.. probably associated with levels of disseminated sulphides and metal(loid)s enriched crusts.
Line 283: The discussion of the IP data has been discussed along with the resistivity in each profile.
Line 301: I have added… Low chargeability values, probably due to the absence of dispersed metals, generate a weak polarization phenomenon, versus to the high values.
Line 311: I have added… (ERT and IP)
Line 318: I have cross out resistivity
Line 324: method
Line 329: methods

Reviewer 2 Report
Dear Editor;
I am sending you my comments on the manuscript entitled “Characterization of tailings dams by electrical geophysical techniques (ERT, IP): Federico Mine (La Carolina, southeastern Spain)”.
The present manuscript is written in a good way but it includes some issues that need to be addressed.
- Some comments relating to the language and references can be found in the attached file.
- Some information are missing or need to be clarified such as:What is the RMS values, is the topography incorporated in the inversion process, some inconsistency in the results of ERT and IP (for example: ERT 6 and IP Federico 6).
- Lines 283 to 309: the text should be written in better way.
- I advise to avoid over-interpretation (refer to Fig. 5 (ERT3).
- A 3D fence diagram combining profiles ERT 1 to 5 would show the subsurface resistivity image more clearly.
More comments can be found in the attached file.
Good luck

Author Response
Response to Reviewer 2 (In color blue)
(Response to comments made in the pdf)
Line 63: I have modified the sentence by crossing out what the referee indicates
Line 64: I have added the references
Line 82: I have modified the sentence
Line 188: in response to referees 1 and 2, I added… in materials and methods.. We have chosen the Wenner-Schlumberger arrangement which has a good penetration capacity and a good sensitivity for both vertical and horizontal structures. Other electrode arrangements such as the Wenner or dipole-dipole configurations are more appropriate for sub-horizontal structures and vertical structures, respectively.
Line 189: I have rewriting the phrase for better explain the distances.
Line 202: I have removed “previously” and I have added… The topographic information has been entered into the model.
Line 213: We indicated more than 1,000
In point 4. Discussion and Results, I have added the RMS
In figure 5, We know from field observation of the existence of the drainage gallery. In Figure 1b, roughly where it runs is indicated (green dashed line). However, I removed the reference to the drain gallery in the ERT-3 profile as it was not well marked. I also modified the caption of the figure.
In figure 6, in IP 5, I have removed the anomaly from the gallery as it is not clear and does not match ERT 5.
With respect to ERT 6 and IP 6, the high resistivity is due to metal-enriched crusts observed on the surface and that give high chargeability.
Line 287-308: IP results have been discussed along with resistivity
Line 379: I have eliminated space
